# Electrical Conductivity, pH, Minerals, and Sensory Evaluation of *Airag* (Fermented Mare’s Milk)

**DOI:** 10.3390/foods9030333

**Published:** 2020-03-12

**Authors:** Ryouta Tsuchiya, Takayuki Kawai, Tserenpurev Bat-Oyun, Masato Shinoda, Yuki Morinaga

**Affiliations:** 1Organization for the Strategic Coordination of Research and Intellectual Properties, Meiji University, 1-9-1 Eifuku, Suginami-Ku, Tokyo 168-8555, Japan; ryouta1004h@yahoo.co.jp; 2Arid Land Research Center, Tottori University, 1390 Hamasaka, Tottori 680-0001, Japan; kawai@alrc.tottori-u.ac.jp; 3Information and Research Institute of Meteorology, Hydrology and Environment, Ulaanbaatar 15160, Mongolia; nuyo792000@yahoo.com; 4Graduate School of Environmental Studies, Nagoya University, D2-1(510) Furo-cho, Chikusa-ku, Nagoya 464-8601, Japan; shinoda.masato@gmail.com; 5School of Commerce, Meiji University, 1-9-1 Eifuku, Suginami-Ku, Tokyo 168-8555, Japan

**Keywords:** *airag* (koumiss), fermented mare’s milk, sensory evaluation, electrical conductivity, pH, macro minerals

## Abstract

Traditional *airag* (fermented mare’s milk) is a sour, slightly alcoholic drink handmade by Mongolian nomads. As *airag* is not heated after production, the fermentation continues to proceed and the taste changes rapidly. The objective of this study was to investigate the association of the sensory taste evaluation of *airag* with some properties—electrical conductivity (EC), pH and concentrations of macro minerals (calcium (Ca), phosphorous (P), sulfur (S), magnesium (Mg), potassium (K), and sodium (Na))—of *airag*. We held an *airag* contest in Mogod county, one of the most famous *airag* production areas, in order to collect samples of *airag* for the analysis of *airag* properties and to conduct an *airag* taste evaluation by Mongolian people. The results of the statistical analysis indicated that the EC-value was related to the evaluation score of *airag*. Except for EC, no statistically significant relationship between the taste score and the other properties was found in this study. It was concluded that the EC-value would be a simple measurement indicator for evaluating the quality of *airag* on site.

## 1. Introduction

*Airag*, also known as *koumiss* or *chigee*, is fermented mare’s milk, which is a sour and slightly sparkling milk beverage with a low-alcohol percentage (normally less than 3%), and has been widely considered to be good for one’s health. It is a traditional beverage of the nomadic people in the steppe region of Eurasia and has been since ancient times. However, today, most *airag* is industrially produced in farms and factories due to the rapid sedentarization that occurred during the 20th century, except for in Mongolia, where the nomadic lifestyle is still active in the countryside. The traditional technique of producing *airag* in a *khokhuur* (cowhide vessel) in Mongolia was inscribed in December 2019 on the Representative List of the Intangible Cultural Heritage of Humanity.

As *airag* is mainly self-produced and self-consumed in Mongolian households, data about *airag* production are inadequate. Therefore, a questionnaire survey on *airag* targeting 2045 herders was carried out in 2012, and the results showed that *airag* is not produced evenly throughout Mongolia, and the production is intensive in Central Mongolia. Furthermore, the main factors influencing the quality of *airag* were labor, starter, skill, container, grass, weather, minerals, mares, and so on [1]. *Airag* is not only used as a food for daily consumption but also as a sacred beverage for celebratory occasions. In addition, it is believed that the intake of *airag* is good for one’s health, and there are many *airag* sanatoria in the Mongolian countryside.

The traditional method for *airag* production is simple but labor-intensive: adding fresh raw mare’s milk into a container with *airag* previously produced as a starter and stirring it by hand a few thousand to 10,000 times a day using a stirring rod, then leaving it overnight to proceed with its fermentation. The next morning, it is ready to be served. This is a general method for *airag* production, but the home-made recipe, such as the timing and amount of adding raw mare’s milk, the amount of stirring, etc., depends on the experiences and the feeling of the producers. They, along with the starter, bring forth variations in the quality of *airag* products from home to home. Some environmental factors, such as temperature, humidity, water resource, vegetation, etc., are also expected to relate to the *airag* quality by affecting the growth and activity of bacteria and yeast and the health of the horses. Furthermore, *airag* is not pasteurized, making it difficult to stabilize the quality after production, often leading to rapid quality deterioration. In Mongolia, along with a rapid increase in urban population in recent years [2], as well as increasing consumer interest in healthy and traditional foods, the demand for traditional *airag* produced in the countryside is increasing, especially in urban areas. It is considered that the distribution of *airag* produced in the countryside to urban areas has increased. Therefore, it is necessary to develop a simple method in order to evaluate the quality of *airag* on site.

Mogod county in Bulgan province, which is located in the northern part of the active *airag* production area and belongs to the forest steppe zone, is a well-known place for producing good-tasting *airag*, and is a highly conscious area for protecting and maintaining traditional *airag* production [3]. In Mogod county, we investigated several factors related to *airag* production: the effects of horse management on the movement of dams [3] and the food habits of horses by fecal analysis [4]. On the other hand, the taste of *airag* itself and its related properties have not been reported. We held the *airag* contest on 4 August 2016 with the purpose of collecting *airag* samples that were evaluated with a taste score by Mongolian people.

The objectives of this study were to investigate the association of the *airag* properties with the evaluation taste score of *airag* produced by the traditional method in Mogod county in Mongolia.

## 2. Materials and Methods

### 2.1. Sensory Taste Evaluation

A total of 51 *airags* were exhibited at the *airag* contest held on 4 August 2016 at Mogod county cultural center. The exhibits (one for each exhibitor) were produced by herders themselves who lived in Mogod county. All exhibits were evaluated on a scale of 1 to 5, respectively, by five Mongolian judges selected by the organizers. Three were herders, two were office workers, and all drank *airag* regularly. As the judges were not experts in *airag* screening, the sensory evaluation was simplified, and the score was determined by personal preference. In this study, a large number of the exhibits were evaluated in a short time, and the evaluation item was limited to tastes that greatly contributed to the *airag* quality. Each exhibit was coded to avoid identifying exhibitors and scored for preference on a five-point scale ranging from “least preferred” (score = 1) to “most preferred” (score = 5). The median value of the scores by the five judges was determined as the evaluation score for each sample.

### 2.2. Sample Collection

To analyze the *airag* properties, *airag* samples (20 mL each) were collected from each exhibit of the *airag* contest. After measuring the values of electrical conductivity (EC) and the pH of the samples on site, they were pasteurized and transported by air to the Arid Land Research Center, Tottori University, Tottori, Japan, at −20 °C until they were used in the experiments.

### 2.3. EC, pH, and Mineral Concentrations

For each *airag* sample, the following parameters were determined: electrical conductivity (EC) using a conductivity meter (HORIBA LAQUAtwin B-771, HORIBA Ltd., Kyoto, Japan), pH using a pH meter (HORIBA LAQUAtwin B-712, HORIBA Ltd., Kyoto, Japan), macro mineral concentrations (Ca, P, S, K, Na, and Mg) by inductively coupled plasma–mass spectrometry (ICP-MS) (Agilent 8900, Agilent Technologies Inc., Santa Clara, CA, USA).

The values of EC and the pH of the samples were measured directly in situ with the measurement meters immediately after sampling. The concentrations of the minerals were measured by ICP-MS after the digestion of the sample with 5 mL nitric acid in closed vessels by a microwave system (ETHOS UP, Milestone General, Kawasaki, Japan) and diluted 200 times; 5 g of the *airag* sample was weighed and added to the digester vessels. Then, 69% nitric acid for ultratrace analysis (specific gravity 1.42, FUJIFILM Wako Pure Chemical Corporation, Osaka, Japan) was added to the vessels and placed in the digester for a programmed period of time with a controlled temperature and pressure. After that, the vessels were cooled for 30 min. The digested samples were diluted by ultrapure water (>18 MΩ cm), obtained by purifying distilled water with the Milli-Q Reference water purification system associated with an Elix Essential UV 10 pre-system (Merck Millipore, Tokyo, Japan).

### 2.4. Statistical Analysis

The association of the *airag* properties with the taste score was analyzed with a classification and regression tree (CART) performed using R (version 3.2.2, R Development Core Team, 2016), using the package “rpart”. After the CART analysis was completed, trees were pruned based on a “1 SE rule” [5]. The other statistical analyses were carried out on a personal computer using the Ekuseru-Toukei 2010 statistical software package (SSRI, Tokyo, Japan). Statistical significance was accepted at a value of *p* < 0.05.

## 3. Results

The taste scores of the 51 exhibits in the *airag* contest were concentrated at three and five points, showing a distribution with two peaks (Figure 1). About 55% (28/51) of all exhibits were scored with four points or more, and about 45% (23/51) were scored with three points or less, and they were considered to be divided into two groups: high-score (≥4 points) and low-score (≤3 points) groups.

The CART analysis for visualizing the association of the *airag* properties with a score of *airag* showed that the *airag* scores were demarcated into two groups at the boundary of the EC-value of 3.35 (mS/cm). The group with an EC-value of less than 3.35 (mS/cm) included many of the four and five-point *airags*, and the group with an EC-value of 3.35 (mS/cm) or more included many of the two and three-point *airags* (Figure 2). Comparing the mean values of the respective properties between the high-score (≥4 points) group and the low-score (≤3 points) group, only the mean EC-values between the two groups were statistically significantly different. The mean EC-value of the high-score group was lower than that of the low-score group (Table 1). Spearman’s rank correlation analysis also showed a statistically significant relationship between the score and the EC-value, but not between the score and the other properties (Table 2). From these results, we considered that relatively high EC-values lowered the taste scores of the *airag*.

## 4. Discussion

Bornaz et al. [6] showed that the fermentation of Arabian mare’s milk with lactic acid bacteria increased EC dramatically. They suggested that the factor of the increase in EC could be explained by the phenomenon in which minerals dissolve as ions due to the demineralization of micelles caused by the acidification of mare’s milk. The acidification is caused by the conversion of lactose to lactate as a result of the metabolism of lactic acid bacteria. Yoshida et al. [7] and Lanzanova et al. [8] measured the change of EC in bovine milk during lactic acid fermentation and concluded that it was useful to evaluate the growth and metabolic activity of lactic acid bacteria in milk. Mukchetti et al. [9] studied the mechanism of EC elevation in milk caused by acidification and found that both the fermentation of lactose to lactic acid and the acidification of milk changes the equilibria of the buffer systems and solubilizes casein-bound calcium, and phosphorus salts cause the elevation of EC. In this study, the mean EC-value of the high-score group was lower than that of the low-score group even though the degree of acidification (the mean pH-value) was similar between the two groups. Although the difference was not statistically significant, both calcium and phosphorus concentrations were slightly lower in the high-score group compared to the low-score group. This may have resulted in a difference in EC-values between the two groups, but requires further studies.

It is known that not only lactic acid bacteria but also many microorganisms, such as yeasts, are involved in *airag* [10,11,12,13,14,15], and both lactic acid bacteria and yeast have stimulatory and inhibitory effects on each other [16]. A change of EC was also observed in yeast cultures [17], and it was used to analyze the metabolic activity of lactic acid bacteria and yeasts during the lactic-acid‒alcohol fermentation of goat’s milk and mixtures of goat’s milk and mare’s milk [18]. Applying the above discussion to this study, the EC-value is considered to indicate the result of microbial metabolism activity.

To the best of our knowledge, this study is the first to investigate a relationship between sensory taste evaluation and the properties of *airag* and the first to determine the relationship of the sensory taste evaluation and the EC-value. Since the EC-value can be easily measured on site, it can be expected to be a useful indicator for controlling the domestic production of *airag*. A further study is also needed on the fermentation and the EC of *airag*.

## Figures and Tables

**Figure 1 foods-09-00333-f001:**
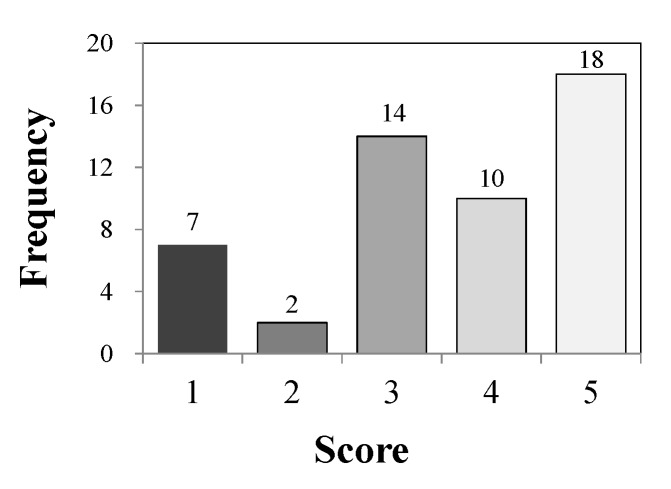
Frequency distribution of the number of the exhibits at each score in the *airag* contest.

**Figure 2 foods-09-00333-f002:**
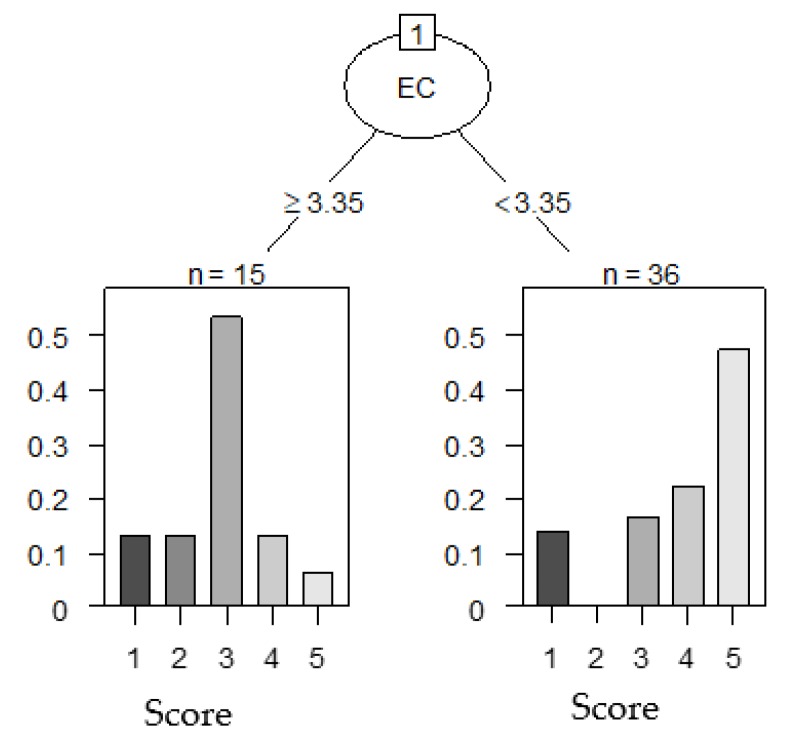
Classification and regression tree (CART) analysis of the *airag* properties (electrical conductivity (EC), pH, concentrations of minerals (Ca, P, S, Mg, K, and Na)) affecting the taste score of *airag*. Trees were pruned based on the complexity parameter corresponding with the smallest cross-validated error.

**Table 1 foods-09-00333-t001:** Comparison of the mean values of the properties of *airag* between high-score (4~5 points) group and low-score (1~3 points) group, and the overall means in the samples.

	*N*		EC *	pH	Ca	P	S	Mg	K	Na
			(mS/cm)		(mg/L)	(mg/L)	(mg/L)	(mg/L)	(mg/L)	(mg/L)
Low-score group	23	Mean	3.23	3.57	740	441	217	53	431	113
(1~3 points)		SD	0.25	0.22	98	101	35	13	92	23
High-score group	28	Mean	3.05	3.53	707	425	217	53	427	114
(4~5 points)		SD	0.37	0.20	101	115	27	13	98	24
*P*-value			*p* < 0.05	NS	NS	NS	NS	NS	NS	NS
Overall	51	Mean	3.13	3.54	722	432	217	53	428	114
		SD	0.32	0.20	100	108	31	13	95	24

EC, electrical conductivity; Ca, calcium; P, phosphorous; S, sulfur; Mg, magnesium; K, potassium; Na, sodium; SD, standard deviation * Significant difference at *p* < 0.05 using a *t*-test; NS, not significant.

**Table 2 foods-09-00333-t002:** Spearman’s rank correlation between the taste score and the properties of *airag*.

	N		EC *	pH	Ca	P	S	Mg	K	Na
Score	51	Spearman’s rho	−0.28	−0.04	−0.21	−0.11	−0.03	−0.05	−0.07	−0.05
		*p*-value	*p* < 0.05	NS	NS	NS	NS	NS	NS	NS

Score, evaluated score of *airag*; EC, electrical conductivity; Ca, calcium; P, phosphorous; S, sulfur; Mg, magnesium; K, potassium; Na, sodium; SD, standard deviation * Significant difference at *p* < 0.05 using a *t*-test; NS, not significant.

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
