# Peer review of "Electrical Conductivity, pH, Minerals, and Sensory Evaluation of Airag (Fermented Mare’s Milk)"

_foods, 2020, doi:10.3390/foods9030333_

Round 1
Reviewer 1 Report
The topic of the manuscript is of interest, particularly considering that data about fermented mare’s milk are still lacking. The manuscript is easy to understand and is well written.
Specific comments
Line 34
sparkling milk beverage with low-alcohol percentage
What is the alcohol content of koumiss?
Line 60-61
Recently, the consumption of airag has been increasing, especially in urban areas, due to the
growing interest in health and the desire to return to tradition.
Have you got literature data to this information?
Line 72-73
The objectives of this study were to investigate the association of the airag properties with the
evaluation taste score of airag produced by the traditional method in Mogod County in Mongolia.
Why have you decided to score only taste of airag?
Line 81
Sensory taste evaluation
It is not clear. Authors have to explain if the samples of airag were tested after thawing or as a fresh airag?
Line 96
sample with nitric acid
What was the amount of nitric acid?
Line 112
Results and Discussion
I suggest to distinguish a part discussion. Now it is not clear.
Author Response
Thank you for your valuable and informative comments to this manuscript.
Please see the attachment.

Reviewer 2 Report
I found this manuscript to be very interesting from a scientific point of view, especially as novel fermented beverages may well become important in markets outside where they are traditionally produced and there is great interest in these 'functional' beverages currently. The use of electrical conductivity as a means of assessing quality may well be a technique of interest. Chen et al., 2016 did a nice study using conductivity measurements comparing 6 different sports drinks which might be of interest to you - but not necessary to cite in manuscript.
The English language is generally very good throughout the manuscript - just a couple of issues need to be addressed:
Line 58: 'due to overheat sterilisation.....' I'm not sure what you mean here - perhaps reword this sentence?
Line 62: change the terms 'has become active' to 'has increased' or 'is increasing'
Line 67: remove the word 'the' before airag
Line 107: insert the word 'using' before R
I agree with you (lines 147-148) that follow up studies would be interesting and necessary. Detailed sensory analysis, saltiness, bitterness etc..would be interesting to correlate with electrical conductivity (again see Chen et al.) for a more complete picture - but again this could be a new study.
Author Response

(The authors gave the same response as above.)
